# Current State of Compassionate Phage Therapy

**DOI:** 10.3390/v11040343

**Published:** 2019-04-12

**Authors:** Shawna McCallin, Jessica C. Sacher, Jan Zheng, Benjamin K. Chan

**Affiliations:** 1Unit of Regenerative Medicine, Department of Musculoskeletal Medicine, Service of Plastic, Reconstructive, & Hand Surgery, University Hospital of Lausanne (CHUV), 1066 Epalignes, Switzerland; 2Swiss Federal Institute of Technology Lausanne (EPFL), 1015 Lausanne, Switzerland; 3Phage Directory, Atlanta, GA 30303, USA; jessica@phage.directory (J.C.S.); jan@phage.directory (J.Z.); 4Yale University, New Haven, CT 06520, USA; b.chan@yale.edu

**Keywords:** bacteriophage therapy, compassionate use, antibiotic resistance

## Abstract

There is a current unmet medical need for the treatment of antibiotic-resistant infections, and in the absence of approved alternatives, some clinicians are turning to empirical ones, such as phage therapy, for compassionate treatment. Phage therapy is ideal for compassionate use due to its long-standing historical use and publications, apparent lack of adverse effects, and solid support by fundamental research. Increased media coverage and peer-reviewed articles have given rise to a more widespread familiarity with its therapeutic potential. However, compassionate phage therapy (cPT) remains limited to a small number of experimental treatment centers or associated with individual physicians and researchers. It is possible, with the creation of guidelines and a greater central coordination, that cPT could reach more of those in need, starting by increasing the availability of phages. Subsequent steps, particularly production and purification, are difficult to scale, and treatment paradigms stand highly variable between cases, or are frequently not reported. This article serves both to synopsize cPT publications to date and to discuss currently available phage sources for cPT. As the antibiotic resistance crisis continues to grow and the future of phage therapy clinical trials remains undetermined, cPT represents a possibility for bridging the gap between current treatment failures and future approved alternatives. Streamlining the process of cPT will help to ensure high quality, therapeutically-beneficial, and safe treatment.

## 1. Introduction

The first documented therapeutic case of harnessing the natural antibacterial mechanism of bacteriophages, or phages, for the treatment of a human bacterial infection predates the discovery of antibiotics by two decades [1]. Phages were used experimentally for the treatment of various bacterial infections throughout the 1920s, including cholera (reviewed in [2]), dysentery [3], and staphylococcal infections [4] to varying degrees of success [5,6]. For these early applications, phages needed to be isolated from environmental sources, cultivated on bacterial hosts, and purified in line with technology at the time. The deemed founder of phage therapy, F. d’Hérelle, had a heavy hand in the spread of phage therapy during these early years, which he encouraged by traveling to different countries, such as the Soviet Union, India, Egypt, and others, bringing with him phages and the knowledge of how to use them against human bacterial infections [2,7,8].

As phages fell to the wayside with the pursuit of antibiotics in Western medicine in the 1940s, Soviet researchers continued phage development at the G. Eliava Institute of Bacteriophages, Microbiology, and Virology in Tbilisi, Georgia [7,9]. There, phages were isolated from environmental sources and accumulated into a phage bank that exists to this day. This collection provides a large repertoire from which phages can either be incorporated into pre-formulated products or selectively matched against bacterial isolates for personalized therapies. As a result of historical clinical trials and experience accrued during the twentieth century, phages exist alongside antibiotics as approved medicines in some former Soviet Union countries. However, historical data from one country holds little scientific weight in present day evaluations of unapproved medicines in others.

Now, the rest of the world has a re-found interest in revitalizing phage therapy that has paralleled the rise of antibiotic resistance [10,11,12,13,14]. For phage therapy to be recognized as an effective alternative to antibiotics, it will require efficacy data from randomized, controlled clinical trials (RCTs). The three phage RCTs completed to date have failed to produce robust conclusions on efficacy, therefore leaving phage therapy in limbo in the approval process until future trials are conducted [15,16,17]. Only one RCT for phage products is currently open for enrollment (ClinicalTrials.gov Identifier: NCT03808103), although several are scheduled for patient enrollment in the near future. In the interim, several competency centers, physicians, and researchers are invoking phage therapy for compassionate means in order to respond to the current clinical needs of patients suffering from antibiotic failure.

## 2. Compassionate Use

Compassionate treatment denotes the use of unapproved medicines outside of clinical trials for the treatment of patients for which approved therapeutic options have been exhausted. The principle of compassionate use is codified in the “Helsinki Declaration of Ethical Principles for Medical Research Involving Human Subjects”, which is an international agreement on facets of clinical research, such as patient consent and placebo control [18]. Article 37 specifically asserts a physician’s authority to act in the best interest of their patient by using experimental treatments in the absence of approved options, although the support of using unproven treatments was not stipulated by the Declaration until its amendment in 2000 (v2000, Article 32) [19]. In its current state, it reads in its entirety, “*In the treatment of an individual patient, where proven interventions do not exist or other known interventions have been ineffective, the physician, after seeking expert advice, with informed consent from the patient or a legally authorised representative, may use an unproven intervention if in the physician’s judgement it offers hope of saving life, re-establishing health or alleviating suffering. This intervention should subsequently be made the object of research, designed to evaluate its safety and efficacy. In all cases, new information must be recorded and, where appropriate, made publicly available*” [18].

The term “compassionate use” can therefore be referred to both vernacularly in this general sense, as well as formally as a regulatory pathway (also referred to as “expanded access” or “special access”). The process and conditions for compassionate use are stipulated by regulatory agencies, such as the Food and Drug Administration (FDA) in the United States, the Therapeutic Goods Administration (TGA) in Australia, or the European Medicines Agency (EMA) in the European Union (EU), although EU member states apply EMA directives independently and may be governed by additional national regulation [20,21,22]. The objective of compassionate treatment differs from RCTs in that its primary aim is to provide therapeutic benefit to the patient, rather than to evaluate the efficacy of the experimental treatment (although safety may be evaluated). While the term “compassionate” is frequently associated with case reports of phage therapy, it does not inherently signify regulatory adherence, and legal processes that are required for compassionate treatment vary from country to country [23].

The general prevalence and importance of compassionate use is changing, with an increase in access requests and legal support [24,25,26]. Instigation of compassionate treatment also increasingly arises from patient advocacy groups or patients via social media platforms, to bring attention to, put pressure on, and finance access to unapproved therapies [27]. “Right-to-try” legislation in the US aims to expedite treatment of severely ill patients with unapproved medicines, albeit with lower regulatory and safety oversight [25]. While the intention is to increase therapeutic options for patients and highlight the inability of current pathways to respond punctually to medical needs, it is not without consequence for ethical considerations, such as equal access, unfulfilled expectations, data collection/usage, or financial responsibility [23,24,25,28].

## 3. Compassionate Phage Therapy (cPT)

The potential utility of cPT is considered after antibiotic failure is clearly documented, attempts to use conventional treatment have been exhausted, and there are no active clinical trials suitable for enrolment (Figure 1). The possibility of using phages may be suggested by the physician, medical entourage, or the patient themselves. Both the consent of the physician and the patient or guardian are essential for continuing the process of cPT, which may or may not be subject to additional institutional or national regulation on the use of unapproved or experimental therapies. cPT has been approved under emergency investigational new drug (eIND) and expanded access schemes by the FDA, a temporary use authorization (ATU) by the French National Agency for Medicines and Health Products Safety (ANSM) in France, by special access schemes by the TGA in Australia, and by national regulation in Poland. Expanded/special access schemes facilitate access to products in clinical development for compassionate treatments and several phage products have fallen under such schemes in the US and Australia [29,30,31]. Without local support, physically- and financially-able patients have the option of traveling to receive phage therapy in countries where it is an approved medicine: For instance, the Eliava Institute in Tbilisi, Georgia has provided treatment to a number of international patients on-site [32,33,34]. The exact process for organizing cPT remains highly variable at present due to its compassionate nature. It can represent a time-consuming endeavor for new cases, to the extent that it may deter motivation to pursue cPT as an option or delay the initiation of treatment, which may influence therapeutic outcomes. Competency centers or individuals experienced with cPT have the advantage of activating familiar pathways for subsequent treatments, and it is the experience of the authors that these centers and individuals are generally willing to be consulted for information on how to best initiate and follow through with cPT. The cost of providing a phage suitable for human application is currently high, with the financial burden falling on the phage provider for most cPT cases, although this may vary between countries and the regulatory status of phage therapy.

Some countries have established legislation for phage therapy without marketing approvals for phage products, such as the Ludwik Hirszfeld Institute of Immunology and Experimental Therapy in Poland, which has been treating patients with phages experimentally with outpatient care since the 1970s. The Phage Therapy Unit (PTU) was opened there in Poland in 2005, which operates phage treatment under a national regulation scheme, and researchers have published summaries and case reports on nearly 1500 patients since 2000 [35,36,37,38,39,40]. Costs for cPT are more realistically managed in Poland, where research institutions, such as the PTU, are not permitted to cover healthcare-related costs, leaving payment to the patients, insurance companies, or sponsors. The Center for Innovative Phage Applications and Therapeutics (IPATH) at the University of California San Diego School of Medicine opened mid-2018 as the only present-day phage center in North America with a clear intention of using phages for compassionate needs and for the eventual elaboration of clinical trials [41]. Experience with several cPT treatments in Belgium led to a recently orchestrated permission to use phages as active ingredients of magistral preparations (known as compounded prescription drugs in the US) [42]. This framework allows phages to be prescribed for individual patients as long as they are produced according to an internal monograph. Phages are still considered “non-authorized” components of the preparation, however, and the availability of magistral phage preparations is still limited, even within Belgium. While this model is distinct from compassionate use, it illustrates how compassionate use can lead to the elaboration of alternative approval pathways with clearly-defined guidelines, even if they are unlikely to be replicated in countries, such as the US, where compounded components require authorizations. Beyond such phage competency centers, unassociated physicians have occasionally independently administered phages from academic, biotech, and commercial sources for the treatment of antibiotic resistant infections [32,43,44,45,46,47].

There are more than 25 reports of cPT since 2000, half of which have been published in the past two years and represent different infections, phages, pathogens, and administration routes that collectively represent the application of phages to nearly 2000 people (Table 1). These case studies are published either as periodic updates on the experiences of competency centers or zealous physicians or researchers, or as isolated one-off applications. They vary widely in the information included within the publication, concerning treatment outcomes, concomitant antibiotic use, and microbiological assessment. Instances of cPT usually incur a lag time to publication or are presented at conferences or published as press releases rather than peer-reviewed publications, meaning that there are more cPT cases occurring than published through scientific channels. Indeed, Ampliphi Biosciences have announced via press-release an 84% clinical success rate through their expanded access programs for the treatment of *Staphylococcus aureus* or *Pseudomonas aeruginosa* infections [30,48].

From published cases, treatment with cPT for *S. aureus* infections has been reported the most frequently, followed by *P. aeruginosa* and *Escherichia coli*, and to a lesser extent, *Enterococcus* sp., and *Acinetobacter baumannii* (Table 1). Cases include the treatment of a myriad of different indications for both chronic and acute conditions, including bone-and-joint, urogenital, respiratory, wound, cardiac, and systemic infections, via various administration routes. Positive treatment outcomes range from 40 to 100% of patients included in reports of more than one participant, depending on the size of the study and heterogeneity of treatment strategies (monotherapy versus cocktail; phage substitution; combination with antibiotics). The development of resistance to applied phages was microbiologically documented in only five reports and largely uninvestigated or unreported in most studies, even in the event of unsatisfactory clinical outcomes. Larger reports show treatment failure rates between 4% and 60%, again with differing methodology between studies with little analytical explanation as to how or why failure occurred. Even definitions of clinical “success” or “failure” may vary, therefore cautioning against the over-interpretation of some cPT results. While publishing cases of cPT helps foster familiarity with phage therapy and support claims of safety, it is not possible to draw conclusions on broader efficacy or to use compassionate treatments in lieu of clinical trials. More standardized reporting guidelines would, however, be useful in order to make comparisons between treatments, particularly in terms for the development of phage resistance (Oechslin and McCallin, submitted).

## 4. Sources and Availability of Phages for cPT

An essential prerequisite for cPT is the availability of phages active against the patient isolate that can then be sufficiently purified to support clinical application. While evident, this can be a limiting factor for cPT considering both the high level of specificity of phage–bacterial interactions and time-to-treatment constraints for acute infections. Possible sources of phages for cPT are summarized below, all of which have contributed by varying extents to cPT efforts.

### 4.1. Environmentally-Sourced Phages

Phages are naturally present in abundance from environmental samples, particularly in bacteria-rich environments, such as sewage or from infections themselves, and natural environments have been the primary source for all phages used in cPT to date [64]. However, starting from this point entails phage isolation, propagation, and characterization that can delay treatment considerably, and requires research infrastructure and expertise. Rare or less-studied pathogens may necessitate environmental isolation of new phages, whereas phages against well-known pathogens (e.g., *S. aureus, P. aeruginosa, E. coli*) are already widely available.

### 4.2. Academically-Sourced Phages

Phages are the subject of fundamental and translational research in numerous academic laboratories around the world. As such, phages sourced from academic labs often offer the benefit of additional characterization, such as genome sequencing, host range analysis, and in vitro/in vivo studies that can provide further information to support their use for cPT. Examples of cPT cases that used phages sourced from academic labs include Schooley et al. [57] and Chan et al. [43]. In addition to academic labs, phages can also be sourced from established phage banks or repositories, some of which provide phages across international borders. Examples of phage banks include the Félix d’Hérelle Reference Center for Bacterial Viruses at the University of Laval [65], the Leibniz Institute DSMZ-German Collection of Microorganisms and Cell Cultures [66] and the Bacteriophage Bank of Korea [67]. Phages sourced from such banks are also often well-characterized, but may incur standard purchasing costs, while academically-sourced phages tend to be supplied pro bono. Large phage banks can provide the benefit of wider pathogen coverage, while some academic phage collections only include phages against one or a select few pathogens. In addition to large phage banks that serve the international community, other phage banks are intended to supply phages for in-house or local cases. For instance, the collection at the PTU contains over 500 phages that cover 15 bacterial pathogens; however, their phages have not been reported for cPT outside of Poland [40]. Both academically- and bank-sourced phages may be liable to intellectual property (IP) constraints, though to different degrees, or require a material transfer agreement (MTA) that limits and delineates the use of the phage(s) supplied.

### 4.3. Phage Products in Clinical Development

Phages are progressively being developed for clinical use by biotech companies. Such companies as Pherecydes Pharma (France), Ampliphi Biosciences (US, Australia), and Adaptive Phage Therapeutics (US) have participated in the supply and preparation of phages for cPT patients [52,53,54,57]. Phages from clinical developers are well-suited for cPT, but phage biotechs understandably retain the right to decline phage supply in consideration of their capacity and business interests.

### 4.4. Eastern European Phage Products

Commercially-available phages and phage preparations from Eastern European countries are an additional phage source that have been used in clinical trials [16] and in compassionate treatments, both within countries where they exist as registered products and in Western countries [32,33,34,45,46,47,51,55]. While standard commercial preparations have a predefined composition of phages, the Eliava Institute offers personalized [34,47] or adapted phage compositions [50] that have been used in Tbilisi or sent to other countries such as France, the US, or Australia. However, the use of commercial preparations from Eastern Europe for cPT in countries where phage therapy is not approved may, or may not, lead to importation or approval difficulties depending on regulatory adherence and requirements.

### 4.5. Crowd-Sourcing Phages

The importance of phages, whatever the source, for cPT is that they have activity against the patient isolate, can be purified and formulated for safe administration, and are readily available to conduce punctual treatment. The need for coordinated phage sharing was documented within a cPT case for the treatment of a multidrug-resistant *A. baumannii* infection with phages [57]. In this case, a total of nine phages from three different sources were required, and the effort was largely coordinated by the patient’s wife via email and social media outlets due to the absence of established or official channels. Following this case, in 2017, an initiative to organize such sharing was founded called Phage Directory (https://phage.directory) [68]. One focus of this initiative is to keep a register of academic phage researchers, phage banks, and phage companies that are willing to contribute phages for cPT in order to locate active phages in the most time-efficient manner. For example, in late 2018, Phage Directory helped coordinate the sourcing of *Klebsiella pneumoniae* phages for a patient in Helsinki, Finland by sending an electronic alert to its network of registered labs and phage collections. This effort resulted in >175 phages being contributed by ten different groups over the span of three weeks, all of which were tested against the patient’s isolate [31]. As of January 2019, there were 36 academic phage laboratories and one phage bank registered on Phage Directory, representing more than 20 different countries with phages covering more than 32 host genera (Figure 2). While this sharing network may be less important for established centers or for those with direct access to large phage collections, it certainly facilitates access for geographically-removed patients or physicians without phage research support or established connections.

### 4.6. Logistical Constraints

Phage sharing still requires the shipping of bacterial strains and/or phages to various locations across the world; often phage biotechs or phage banks prefer bacterial strains to be shipped to them for sensitivity testing, whereas academics have been more willing to send phages directly to other researchers. From a regulatory point of view, shipping phages does not raise biosafety concerns. However, shipping pathogenic bacteria does and is subject to pathogen transport regulations regarding labeling, packaging, and documentation. In either situation, this step for cPT is time-consuming and expensive, and represents a point of intervention for simplifying cPT. The ability to centralize stock of phages available for cPT or to perform on-site susceptibility testing would reduce costs, standardize susceptibility testing, and reduce time-to-treatment for cPT and clinical trials alike.

## 5. Beyond Availability

Active phages are indeed indispensable for cPT, but several subsequent considerations need to be addressed in order to assure a sound therapy. How phages are transported, amplified, purified, and formulated into clinically-applicable formulations remains variable between cPT cases. These processes require the oversight of, and close collaboration between, competent phage scientists and physicians. Disorganization is a risk factor for errors to arise throughout this process, and measures must be taken so as not to compromise to the integrity of the phage product and subsequent therapy. Verifying phage viability, compatibility with medical devices (such as tubing or nebulizers), and sustained activity against a patient’s infection throughout treatment are not systematically included for cPT, although they are important factors for achieving intended therapeutic benefits. Compassionate use is not subject to consistent procedures, and the employment of non-standardized methods for sensitivity testing, purification, or formulation could contribute to variable treatment outcomes that may be difficult to explain without thorough analysis. Phage therapy walks a thin wire for retaining support to becoming part of modern medicine due to the historical hangover of inconsistent observations in early trials, which continues to cast doubt on the potential of phage therapy today (reviewed in [69,70,71]). Another risk for cPT treatment is that candidate patients often have confounding medical conditions that complicate prognosis, although phage administration has never been linked to cause of mortality [44,56]. These considerations make the effort to ensure that cPT is consistent and cooperative all the more important.

However well-coordinated these processes become, the cost of providing cPT treatment is a constraint on its scalability. Financial estimates for production costs and manpower needed on a per-case basis are difficult to come by, but have ranged in the tens of thousands of US dollars in countries where phage therapy does not have a legal framework (personal communications). cPT is currently provided at no cost to the patient or treating institution in cases of cPT in the US, France, or Australia; a model with little financial viability for either small biotechs or research labs. However, cPT does not represent an avenue for commercialization. As clinical trials open, it is thought that more patients will be able to access treatments through expanded access schemes or even through participation in ongoing trials. The most scalable option is indeed to obtain marketing authorizations for phage products, which, in a catch-22 situation, does little to address the issue of the current medical need for cPT now.

On a final note, inconsistent, incomplete, or a lack of cPT reporting altogether is a missed opportunity for gaining a better understanding of the antibacterial activity of phages in humans and for further developing human phage therapy. The last phrase of Article 37 iterates the importance of recording information gleaned from compassionate use cases and making it publicly available [18]. However, cPT reporting is frequently neglected or delayed for long periods of time following treatment, with traditional news and social media-based reporting often outpacing peer-reviewed publications. Data gathering has been identified as a problem with compassionate programs [72], which is further complicated when compassionate treatment is provided by multiple sources, as in the case with cPT, instead of a singular manufacturer. A better-structured, data-supported coordination of cPT would enable this treatment option to not only become more widespread and ensure safer practices for patients, but also to provide invaluable information to help refine future phage treatments. The focus of compassionate treatment is unquestionably to provide benefit to the patient, but in consideration of the higher success rate with cPT compared to meager RCT results, it is both wasteful and borderline unethical to not thoroughly record and analyze non-efficacy data from cPT cases, such as doses, frequency, or changes to phage sensitivity profiles. Information including pharmacokinetics, concomitant treatment with antibiotics, and the apparition of phage-resistant variants from cPT would be extremely useful in shaping future phage therapy endeavors and avoiding the clinical futility that has been associated with recent phage RCTs. A detailed set of suggested criteria that phage research and therapy should report has been proposed by Abedon [73]. Here we have presented several generalities that should be addressed for cPT, which then next requires a practical proposal to be formulated, supported, and adhered to by multiple stakeholders for the creation of clear policy and actual implementation.

## 6. Conclusions

The duration of time until approved alternatives to antibiotics become available is unreassuringly unknown. Traditional drug development pipelines estimate four to ten years for widespread marketing and distribution of any new medicine or therapy, leaving approved phage products something for the future. This substantial lag time between current need and the earliest foreseeable approvals for new antibacterials leaves a considerable number of patients in a highly precarious situation: Reports estimate that approximately 700,000 deaths are caused by antibiotic resistance each year already [74], and claim an even higher number of disability-adjusted life-years and financial burden [75]. The success rates of the cPT cases that have been reported on to date, as well as the willingness of the phage community to participate in cPT efforts for critically-ill patients, emphasizes the potential role that cPT could play in filling this gap between faltering antibiotics and the development of viable alternatives. However, the case reports of cPT over the past decade have addressed only a negligible proportion of antibiotic-resistant cases and remain geographically concentrated around experimental centers or related to a small number of physicians and researchers with the required know-how. The most impactful way to address antibiotic resistance would be to generate efficacy data through clinical trials that would lead to marketing approvals. In the meantime, with better organizing of cPT in terms of phage availability, logistics, and data reporting, progress can be made in the here and now toward alleviating clinical failures due to antibiotic resistance.

## Figures and Tables

**Figure 1 viruses-11-00343-f001:**
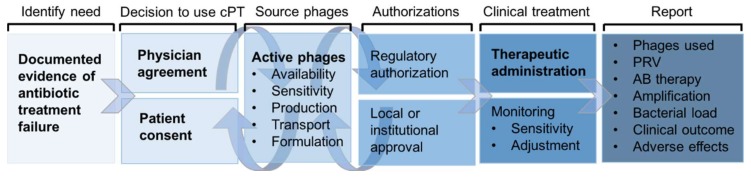
General process and considerations for compassionate phage therapy (cPT). Required steps are shown in bold. Circular arrows indicate processes that are dynamic and do not occur necessarily in a chronological order. PRV: Phage-resistant variant. AB: Antibiotic.

**Figure 2 viruses-11-00343-f002:**
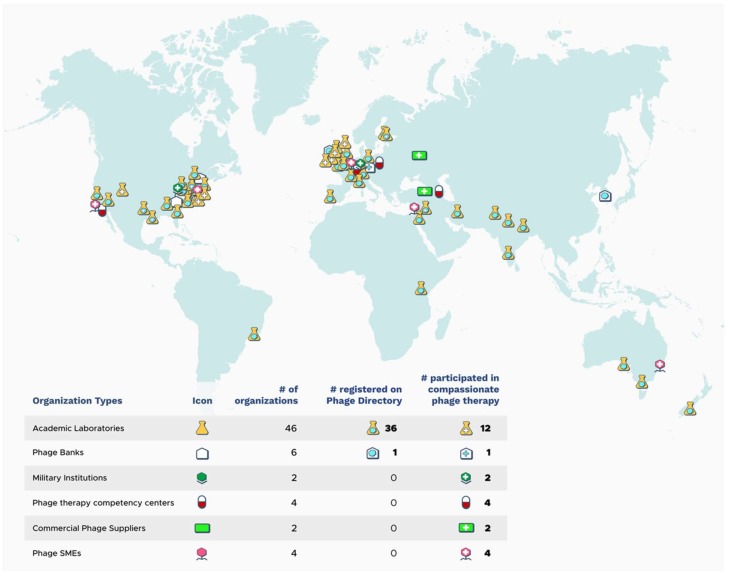
Geographic distribution of organizations (grouped by type) that have either previously participated in cPT cases or demonstrated intent to do so in the future through registration with Phage Directory (numbers current as of January 2019). Phage organizations not having yet contributed to cPT are not listed here. SME: Small- and medium-sized enterprises.

**Table 1 viruses-11-00343-t001:** Summary of 29 publicly-available, published cases of cPT as of April 2019 in chronological order of most recent publication. Causative pathogens, types of infections (mono/polymicrobial; clinical indication), and administration routes vary between studies. The definition of success may be specific to authors, but overall indicates observed clinical amelioration and/or pathogen clearance. Concomitant antibiotic therapy is indicated for the number of patients per study if ≥1. Plausible reasons for cPT failure are listed when available, as well as the investigation into bacterial development of resistance to applied phages. Phage sources used for treatment are listed and further information can be found in cited references.

Pathogen	Infection	Admin Route	N*	Clinical Outcome	AB (N*)	Failure /PRV^+^	Phage Source	Ref.
*A. baumannii, K. pneumoniae*	Bone	iv	1	Success	Yes	na/no	Military	[49]
*S. aureus;* *P. aeruginosa;* *E. coli; Proteus^PM^*	Bone; GI; ENT; urogenital	Local; oral; rectal; joint injection	15	High success rate (12/15); all cases improved	Yes	2° pathogen for 1 patient; unclear results for 2 patients	Mostly commercial	[33]
*S. aureus*	Bone	Soft-tissue injection	1	Success	Int.	na/nr	Commercial (Eliava)	[45]
*S. aureus; E. coli; Proteus; Streptococcus; P. aeruginosa*	UTI	Local via catheter	9	Bacterial load decrease in 67% (6/9); pathogen clearance for 3 patients	Yes (1)	No decrease for 1 patient; 2° infection for 1 patient/ nr	Commercial (Eliava); adapted to strains	[50]
*Achromabacter xylosoxidans*	Cystic Fibrosis infection	Inhaled; oral	1	Improved lung function and general condition	Yes, post	na/nr	Environ.	[51]
*P. aeruginosa*	Recurrent pneumonia	Inhaled; iv	1	Success	Yes	na/ Yes (PS)	Environ., biotech; military	[52]
*S. aureus*	Bone	Local	1	Success	Yes	na/nr	Biotech	
*S. aureus, P. aeruginosa^PM^*	Bone	Local	3	nr	nr	nr/nr	Biotech	[53]
*P. aeruginosa*	Bone	Local	1	Success for bacterial clearance^†^	Yes	na/nr	Biotech	[54]
*E. coli; Proteus; S. aureus; P. aeruginosa; Streptococcus; Enterococcus*	Burns, ulcers, wounds	Topical; sc	234: (27; 90; 94; 23)	Overall high success rate; varied by study	Varied with study	Varied with study/nr	Commercial; unspecified	Review of 4 cases in Russian [55]
*P. aeruginosa*	Aortic valve graft	Direct via fistula	1	Success	Yes	na/nr	Academic	[43]
*A. baumanii*	Post-operative cranial infection	iv	1	Infection site cleared; blood cultures negative^†^	No	Treatment discontinued/nr	Military	[56]
*S. aureus*	Chronic skin infection	Topical; oral	1	Decreased bacterial load; improved clinical condition	No	Prolonged treatment/Yes (PS)	Commercial (Eliava)	[34]
*A. baumanii*	Necrotizing pancreatitis	iv; local	1	Success	Yes	na/Yes (PS)	Environ., military; biotech; phage bank	[57]
*P. aeruginosa*	Infected wound/ septicemia	iv; local	1	Wounds remained colonized, blood cultures were negative^†^	Int	Bacteremia resolved, but local infection persisted/nr	Military	[58]
*P. aeruginosa*	Bacteremia	iv	1	Bacteremia eradicated twice; subsequent regrowth ^†^	Yes	Slow bacterial regrowth/PRV likely	Military	[44]
*S. aureus*	Diabetic toe ulcer infection	Topical	6	Success; avoided amputation	nr	na/nr	Commercial (Eliava)	[59]
*S. aureus*	Corneal abscess	Topical, nasal, iv	1	Success	nr	na/nr	Commercial (Eliava)	[32]
*P. aeruginosa; S. aureus^PM^*	Burn wound infections	Topical	9	Modest reduction in bacterial load for 8 patients	Just prior	nr/nr	Military	[60]
*Staphylococcus; Enterococcus; Pseudomonas; E. coli; Proteus; Enterococcus;* etc*^PM^*	UTI; urogenital; soft tissue; skin; orthopedic; respiratory; bacteremia; etc.	Topical, oral, rectal, vaginal, inhaled	157	Good clinical outcomes for 44% of patients (success for 18%)	Yes (29%)	Inadequate response for 60% of patients/Yes	In-house	[40]
*P. aeruginosa*	UTI	Local in bladder	1	Success	Yes	na/No	Commercial (Eliava)	[47]
*Enterococcus faecalis*	Prostatitis	Rectal	3	Success	No	na/nr	In-house	[36]
*S. aureus*	GI Carrier status	Oral	1	Success	No	na/nr	In-house	[35]
*P. aeruginosa*	Burn wound	Topical	1	Successful grafting	Yes	na/nr	Academic	[61]
*S. aureus*	Wounds	Topical	2	Success	Yes	na/nr	Commercial	[62]
*S. aureus;* *E. coli;* *P. aeruginosa; Klebsiella;* etc*^PM^*	Septicemia	Oral	94	85% success rate	Yes (*n* = 71)	Phage ineffective for 15% of patients/nr	In-house	[39]
*Staphylococcus; E. coli; Proteus; Streptococcus; P. aeruginosa ^PM^*	Venous ulcers and wounds	Topical	96	70% healing	Yes	No clinical improvement for 5 patients	Commercial	[63]
*S. aureus; E. coli; P. aeruginosa; Klebsiella ^PM^*	Various infections in cancer patients	Oral, local	20	Healing in all patients	nr	na/nr	In-house	[38]
*S. aureus; E. coli; Proteus; P. aeruginosa; Klebsiella; Enterobacter^PM^*	Septicemia; ENT; UTI; meningitis; respiratory; wounds; bone; etc.	Oral; topical; local	1307	Full recovery 86%; 11% transient improvement	nr	No effect in 3.8% of study population (*n* = 50)	In-house	[37]

* Number patients in study; AB: Concomitant antibiotic treatment with number of patients in (); PRV: Phage-resistant variants reported; *^PM^* includes polymicrobial infections; GI: Gastrointestinal; ENT: Ear Nose Throat; 2° Secondary; iv: intravenous; Int: intermittent; **^†^** Deceased; PS: Phage Substitution; sc: subcutaneous; na: not applicable; nr: not reported; Environ: Environmental.

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
