# Peer review of "Current State of Compassionate Phage Therapy"

_viruses, 2019, doi:10.3390/v11040343_

Round 1

Reviewer 1 Report

The authors present an interesting and well-written review of application of phage therapy under compassionate use (cPT), including, among other things, a list of the publicly-available published cases of cPT and a brief presentation of the Phage Directory network.  The authors issued also some general recommendations about cPT good practices to be implemented in the future, in line with the provisions of the Declaration of Helsinki.  Still, the proposed recommendations remain at the level of broad generalities and whereas the authors call for the creation of guidelines and process streamlining, apart from a cross reference to a set of criteria for phage therapy report, no practical proposal of implementation was issued.  Nonetheless, the submitted report is of valuable interest and undoubtedly deserves a publication in Viruses, assuming this could be the starting point of further reflection facilitating the expected issue of practical guidance in the future.

The few following points should be reviewed by the authors.

Minor comments

-        Line 134: for the sake of clarity, it is recommended to relate (if applicable) the PTU to the former mentioned Ludwik Hirszfeld Institute.  If not applicable, it should at least be mentioned that the PTU is located on Poland.

-        Line 209: there is typo.  “S. aureus” should be read instead of “S. s aureus”

-        Lines 430-435, references 29 and 30.  Links towards relevant webpages (if any) should be included for these two references.

-        Lines 463, 536, 538, 541 and 542. The transcription of the dates (“January 5th 2019 versus “Jan 05 2019”) should be consistent.

Major comments

-        Lines 81 & 82:  in reviewer’s opinion, the reference to hospital exemption should be removed since this regulatory pathway does not really match with the compassionate use concept, for the following reasons:

  1.       The hospital exemption is restricted to a well-defined and limited category of medicinal products, ie. the ATMPs.  This pathway is thus not applicable to any medicine and definitely not to phage therapy.

2.       As clearly mentioned in the Art 37 of the Helsinki Declaration, behind “compassionate use” is the idea that such interventions are intended to “offers hope of 76 saving life, re-establishing health or alleviating suffering”.  There is no requirement for addressing an unmet need in the hospital exemption pathway for ATMPs.

3.       The hospital exemption must be authorised by the competent authority of a EU Member State, whereas compassionate use doesn’t necessarily, and likely not in the spirit of the above mentioned article of the declaration of Helsinki.

It is thus recommended to remove the reference to hospital exemption.  Instead, the authors may possibly consider introducing other terminologies such as “Special Access Scheme” (used by TGA), “Special Access Programme” (used by Health Canada) etc…

-        Lines 276 – 283: Logistical constraints

From a regulatory point of view, shipping phages is extremely different from shipping bacterial strains.  Whereas the former do not raise biosafety concern, the latter do.  Indeed, bacterial supporting phage growth may be pathogenic and as such, submitted to the biosafety regulation on transport of pathogens (see https://www.biosafety.be/content/safety-measures-transport-gmos-andor-pathogens).  An extensive review of these provisions if of course not the subject of the publication, but the authors should at least emphasize that the shipping of pathogens may entail quite significant regulatory and administrative burden, and should possibly briefly mention some of the international regulations the shipping operations are subject to.

Author Response

Dear Reviewers,

We greatly appreciate the proposed suggestions and have made the corresponding modifications. As the line numbering has changed, we address these changes point-by-point here, with new line numbering when useful:

Minor revisions:

·      It was made clear that the PTU is in Poland: line 137

·      The typo for the bacterial name S. aureus was corrected: line 216

·      Weblinks were added for the two Amplify references 29 & 30

·      The format for the access dates within the references was made consistent

·      Corrections to references Letkiewicz et al. 2010 and Leszczyński et al. 2006 were made

More important revisions:

·      Hospital exemption: This was changed to “special access” and hosptial exemption was removed; we thank the reviewer for their explination on the specifics of hospital exemption: line 81, 113

·      Pathogen shipping: Details to the procedures of shipping phages/bacteria were added; line 289

·      Payment in Poland: The system in for covering the cost of cPT in Poland is more more realistic than that followed in others (US, France), where the cost of recent cases of cPT have been born by the phage manufacturers, primarily. We specified payment in Poland at line 139, where the PTU is described, but did not include this information at line 128 in the generalist description. It is not clear if a for-profit phage biotech would be allowed to accept payment for a phage product (Pherecydes and AmpliPhi provide phage for free), whereas accepting payment may be allowed if produced by a research institution, such as the PTU.

We also took into consideration the comment that no practical proposal was formulated and added a sentence to indicate a next step forward. Indeed, the translation of such review articles, as ours and multiple others, into eventual policy and action is lacking in the phage therapy field. We hope to find an outlet to initiate and accomplish this in the near future.

Lastly, one additional case report was added to Table 1 of a cPT case for the treatment of a polymicrobial bone infection in Isreal (Nir-Paz et al., 2019).

On behalf of myself and co-authors, we greatly appreciate your time and review of our article.

Cordially,

Shawna McCallin

Reviewer 2 Report

line 124: cost of cPT falls on phage provider in most cases - today most cases of cPT come from Poland where someone must cover the costs of phage preparations (a research institution is not permitted by law to cover healthcare-related costs). Therefore, patients, insurance companies or sponsors must cover those costs;

reference 35 : please correct the author list and the name of the journal (Folia Microbiol 2006,51,336-8)

reference 36 Letkiewicz et al: please replace with Letkiewicz et al FEMS Immunol Med Microbiol 2010

Author Response

(The authors gave the same response as above.)
